# Inhibition of Phosphodiesterase-4 in Psoriatic Arthritis and Inflammatory Bowel Diseases

**DOI:** 10.3390/ijms22052638

**Published:** 2021-03-05

**Authors:** Andrea Picchianti-Diamanti, Francesca Romana Spinelli, Maria Manuela Rosado, Fabrizio Conti, Bruno Laganà

**Affiliations:** 1Department of Clinical and Molecular Medicine, S. Andrea University Hospital, “Sapienza” University, 00189 Rome, Italy; bruno.lagana@uniroma1.it; 2Reumatologia, Dipartimento di Scienze Cliniche Internistiche, Anestesiologiche e Cardiovascolari, “Sapienza” Università di Roma, 00161 Rome, Italy; francesca.spinelli@uniroma1.it (F.R.S.); fabrizio.conti@uniroma1.it (F.C.); 3Consultant in Immunology, 00125 Rome, Italy; mariamanuelamrosado@gmail.com

**Keywords:** phosphodiesterase-4, cAMP, enteropathic spondyloarthritis, psoriatic arthritis, inflammatory bowel diseases, ulcerative colitis, apremilast

## Abstract

Phosphodiesterases (PDEs) are a heterogeneous superfamily of enzymes which catalyze the degradation of the intracellular second messengers cyclic adenosine monophosphate (cAMP) and cyclic guanosine monophosphate (cGMP). Among PDEs, PDE4 is the most widely studied and characterized isoenzyme. PDE4 blocking can lead to increased levels of intracellular cAMP, which results in down-regulation of inflammatory responses by reducing the expression of tumor necrosis factor (TNF), interleukin (IL)-23, IL-17, interferon-γ, while increasing regulatory cytokines, such as IL-10. Therefore, PDE4 has been explored as a therapeutic target for the treatment of different chronic inflammatory conditions such as psoriatic arthritis (PsA) and inflammatory bowel disease (IBD). PsA shares clinical, genetic, and pathogenic features with IBD such as ulcerative colitis (UC) and Crohn’s disease (CD), and enteropathic spondyloarthritis (eSpA) represent a frequent clinical evidence of the overlap between gut and joint diseases. Current therapeutic options in PsA patients and underlying UC are limited to synthetic immunosuppressants and anti-TNF. Apremilast is an oral PDE4 inhibitor approved for the treatment of active PsA patients with inadequate response to synthetic immunosuppressants. The efficacy and a good safety profile observed in randomized clinical trials with apremilast in PsA patients have been confirmed by few studies in a real-life scenario. In addition, apremilast led to significant improvement in clinical and endoscopic features in UC patients in a phase II RCT. By now there are no available data regarding its role in eSpA patients. In view of the above, the use of apremilast in eSpA patients is a route that deserves to be deepened.

## 1. Introduction

Phosphodiesterases (PDEs) are a heterogeneous and large family of enzymes, first described about 60 years ago by Ashman et al. [1], which catalyze the degradation of the intracellular second messengers cyclic adenosine monophosphate (cAMP) and cyclic guanosine monophosphate (cGMP). Several isoforms of PDEs have been described encompassing, by now, 11 isoenzyme groups and 50 isoforms [2]. The nomenclature of these gene families has been standardized [3] and classified according to their functional characteristic such as affinities for cAMP and cGMP, inhibitor sensitivities, responses to specific effectors and mechanisms of regulation [4]. cAMP and cGMP modulate several intracellular signal transduction pathways, thus playing a pivotal role in the regulation of different physiologic processes including apoptosis, cell proliferation, inflammation, immune response, and bone remodeling [4]. Conversely, the inhibition of cAMP and cGMP biological effects by PDEs seems to be a pathogenic mechanism involved in the onset and maintenance of different pathologies including chronic obstructive pulmonary disorder (COPD), depression, diabetes, erectile dysfunction, inflammatory bowel diseases (IBD), and psoriatic arthritis (PsA). 

IBD and PsA are chronic, immune mediated diseases that can lead to reduce quality of life and shorten life expectancy, if not timely and adequately treated. IBD include Crohn’s disease (CD) and ulcerative colitis (UC) which may onset at any age, with a peak of incidence between 20–40 and 50–60 years, and a prevalence in Western countries of 300/100.000 subjects [5]. PsA belongs to the heterogeneous group of spondyloarthritis (SpA), it can manifest at any age, with a predominant onset in the late third decade, affecting men and women equally, with a worldwide estimated prevalence around 1% [6]. 

Enteropathic SpA (eSpA) represent the clinical evidence of the bidirectional relationship between gut and joint diseases. This association is quite common; indeed, arthritis is the most frequent extra-intestinal manifestation in patients with IBD, and, on the other hand, up to 60% of patients with SpA has subclinical gut inflammation [7,8,9,10]. SpA and IBD are closely interconnected, sharing some clinical features, genetic predisposition (HLA B27) [11,12] and dysregulation of immunologic pathways and inflammatory cytokines such as the IL-23/IL-17 axis and the TNF. In particular, IL-23 is responsible for the activation of T and of the “so called” type 3 innate cells (ILC3) to produce IL-17 and IL-22. Both, IBD and SpA patients often showed, in the peripheral blood, an increased frequency of Th-17 and ILC3 cells that are frequently found also in peripheral joints, axial skeleton and on the cutaneous lesions of psoriatic patients [13,14] ILC3 cells are critical for the tight junction formation and proliferation of skin and mucosal epithelial cells but once out of the mucosal microenvironment cause a noxious pro-inflammatory milieu [15]. The reasons why these cells traffic from the gut to the articulations are not well defined but one possibility is that bacterial products such LPS cross the epithelial barrier and accumulates in the joints. In addition, breaches in the intestinal mucosa could lead to a disruption of the basal membrane, hyperplasia of goblet cells, activation of Paneth cells, with subsequent increased bacterial translocation and susceptibility to colitis or pathogen infection [13]. The dysfunction of the gut epithelial mucosa can have downstream effect also on the microbiota composition and several shifts in the gut microbiota composition have been reported both in patients with IBD, SpA and psoriasis [16,17,18,19,20]. The increasing knowledge of the “skin–gut–joint axis”, the frequency and complexity of this clinical overlap has pushed to develop multidisciplinary approaches coordinated by both the rheumatologist, gastroenterologist, and dermatologist, in order to improve the diagnostic and therapeutic management. 

The close pathophysiology of IBD and SpA is also demonstrated by the use of conventional targeted synthetic, and biologic immunosuppressive drugs with common mechanism of action. Separate international recommendations are available for the treatment of SpA and UC patients [21,22]. However, when IBD and SpA coexist, the therapeutic strategy should be tailored, according to the variable features of IBD and the clinical manifestations of SpA [8]. In particular, the correct treatment choice should be driven by the predominantly active disease (SpA or IBD) and its clinical subtype (i.e., peripheral/axial; CD/UC, disease extension). Consequently, in patients with active SpA and IBD in remission, the drug has to be selected following rheumatologic recommendations. Last EULAR recommendations involves the use of conventional synthetic Disease Modifying Anti-Rheumatic Drugs (csDMARDs) (i.e., methotrexate, sulphasalazine) as first line therapy for active peripheral SpA. Biological DMARDs (bDMARDs), such as anti-TNF agents, the anti-IL-17A secukinumab and ixekizumab, and the IL-12/IL-23 inhibitor ustekinumab, as well as targeted synthetic (ts)DMARDs, such as JAK inhibitors and the PDE4 inhibitor apremilast, can be adopted in patients inadequately responder to csDMARDs [21]. However, the therapeutic options are more limited in patients with eSpA. Indeed, the pegylated anti-TNF agent certolizumab and ustekinumab are scheduled in CD but not in UC, whereas etanercept and anti-IL-17A agents are not indicated in IBD patients. Finally, the only approved JAK inhibitor for both UC and PsA tofacitinib, has not yet the redeemability in Italy.

Apremilast is an oral PDE4 inhibitor which modulates different inflammatory mediators [23]. It has shown efficacy and a good safety profile in different randomized clinical trials (RCTs) in PsA patients [24,25,26,27]; thus, it has been approved for the treatment of active PsA in subjects who cannot take or have responded inadequately to synthetic immunosuppressive agents [28]. Apremilast has been studied also in one RCT in active UC patients; although, the primary endpoint of clinical remission was not met, the drug determined a significant improvement in clinical and endoscopic features, and markers of inflammation [29].

By critically reviewing the mechanism of action and the most recent data on the literature, here we provide a review of the pathogenic role of PDE4 in PsA and IBD, pointing out the therapeutic potential of PDE4 inhibition in these autoimmune chronic inflammatory diseases. 

## 2. Literature Search

We performed a search of the PubMed database including several interrelated queries (e.g., “phodiesterases” OR “PDE” AND “inflammation” OR “autoimmunity”; “apremilast” OR “PDE4 inhibitors” AND “psoriatic arthritis” OR “inflammatory bowel diseases” OR “ulcerative colitis” OR “enteropathic spondyloarthritis”). We selected only English language publications; restrictions in terms of time were not applied. The search is updated on 23 February, 2021. Given the narrative nature of the review, the articles retrieved were chosen according to their relevance, as judged by the authors. The search results were then supplemented by browsing the reference lists of identified articles, and by including other documents suggested by authors’ experience. 

### 2.1. Pde4 Inflammation and Autoimmunity

As previously mentioned, cAMP is a key element of the signal transduction in various receptors, including T-cell receptor (TCR) [30]. The sole engagement of TCR, by MHC-peptide, raises the levels of cAMP but does not lead to full T-cell activation. In fact, the increased intracellular cAMP, through the induction of the cAMP-protein kinase A (PKA)-Csk, exerts a negative feedback loop that is implicated in the induction of T cell anergy, a nonresponsive state that occurs when T cells are stimulated through TCR/CD3 in the absence of co-stimulation [31,32]. Conversely, to achieve T-cell activation, CD28 needs to be recruited to the lipid raft where it acts to increase raft-localized PDE4 enzymes that enhance the degradation of cAMP and, consequently stimulate T cell activation [33]. 

In addition to PDE4, a role of PDE7 on T cell activation has been demonstrated by Li et al. that upon PDE7 blockade, with an antisense oligonucleotide, were able to inhibit T cell proliferation, and were able to rescue activation by adding a cAMP analog that blocks downstream cAMP signaling [34]. 

PDE7 and PDE4 are the member of PDE family mainly expressed in both CD4 and CD8 T cells [28,35]. PDE7 and PDE4 are both expressed by regulatory T cells (Tregs), and inhibition of both PDE determine an increase in Foxp3 expression [36], but also an increase of Tregs size, enhancing their regulatory effects on conventional T cells. Thus, PDE4/7 inhibition has a dual effect: they alter the activity of T cells by directly interfering with activation signals, but they also increase the suppressing role of Tregs. In neutrophils PDE4 induces chemotaxis, degranulation, and adhesion to endothelial cells through the release of leukotriene B4, IL-8, and adhesion molecules, such as the 2-integrin Mac-1 [37,38,39]. PDE4 inhibitors are also considered potential suppressors of NF-κB-dependent inflammatory conditions [40]. NF-κB/Rel is a family of transcription factors that regulates the expression of many inducible genes including the one for the TNF. cAMP interferes with NF-κB signaling, high intracellular cAMP inhibits NF-κB function; therefore, by controlling the concentration of cAMP, PDE4 might indirectly influence the activation of NF-κB signaling towards an inflammatory or anti-inflammatory condition. In fact, inhibition of PDE4 decreases NF-κB-mediated TNF expression and increases the synthesis of the anti-inflammatory cytokine IL-10 [41] through PKA activation. Notably, the accessibility of cytokine transcription depends upon the presence of multiple cAMP responsive elements (CREs) at the gene promoters. Thus, PDE4 inhibitors may act either negatively or positively on cytokine expression depending on different CRE elements constituting the gene promoters [42]. This is critical for innate immune cells such as macrophages and monocytes, major players sustaining pro-inflammatory cytokine production. Likewise, T cells are also susceptible to inhibition of NF-κB by anti-PDE4 agents. PDE4 seems to control the release of TNF, IL-2, IL-4, and IL-5, but also the expansion of antigen-driven T cell proliferation, as demonstrated by the PDE4 inhibitor, rolipram, in cell cultures, and in animal models [43]. Rolipram exerts a stronger inhibitory effect on Th2 cells than on Th1, either directly or indirectly, by promoting suppression function of Treg cells through the raise of cAMP levels in the microenvironment. This has been clearly demonstrated in vitro and in mouse models of airway inflammation [44]. Th17 cells contribute to the positive loop of inflammation by secreting IL-17A and IL-17F, thus increasing the recruitment of neutrophils into the inflammation sites. PDE4 inhibitors may break this self-sustained cycle of inflammation by reducing IL-17 production by Th17 cells [45], impairing chemotaxis and degranulation of neutrophils and eosinophils and by modulating adhesion molecules on endothelial cells. In non-immune cells, PDE4 inhibitors play an anti-angiogenic role by downregulating E-selectin expression on endothelial cells and decreasing Vascular Endothelial Growth Factor (VEGF)-induced endothelial cell migration [46,47,48].

Thus, acting on several signal pathways of innate and adaptive immunity and in different cell types, PDE4 inhibitors may have a role, in immune-mediated inflammatory diseases. 

### 2.2. Preclinical and In Vitro Data of Apremilast

The rationale for PDE4 inhibition is based on the effect regulating intracellular cAMP concentration on the balance between pro and anti-inflammatory mediators involved in many aspects of the immunopathogenesis and clinical expression of PsA and UC (Figure 1).

In vivo studies on mice models of colitis showed beneficial effects of PDE4 inhibitors, such as rolipram, mesopram, roflumilast, and tetomilast; however, these agents were not furtherly developed because of severe side effects [49].

Gordon et al., demonstrated that apremilast significantly reduced both TNF and MMP-3 production by gut lamina propria mononuclear cells isolated from patients with IBD [50]. More recently, Li et al. analysed the possible protective effect of apremilast in intestinal inflammation and disease severity of dextran sulfate sodium-induced murine UC. Apremilast-treated colitic mice showed a significant improvement in clinical components of UC (weight loss, stool consistency, and rectal bleeding) as well as reduced histological damage [46]. Apremilast could also modulate mucosal immunity in different ways; in particular, it decreased inflammatory cytokines such as TNF, IFN-γ, IL-1β, IL-6, and IL-17A, both in the serum and supernatant of full-thickness colon tissue cultures. Moreover, apremilast decreased the expression of inflammatory chemokines and chemokine receptors, thus blocking infiltration of immune cells into inflamed tissues and preserving epithelial barrier function, as confirmed by in vivo imaging with FITC-dextran. Finally, concurrent to its effect on PDE4 apremilast exerts a suppressive effect on the phosphorylation of NF-κB, STAT1, STAT3, STAT5, and STAT6 via increasing the Suppressor of Cytokine Signaling (SOCS) 3 expression [51]. In lipopolysaccharide-stimulated PBMCs, apremilast increased IL-10 while decreasing TNF, IFN-γ, IL-12, and IL-23 [23].

The effect of apremilast on pro-inflammatory cytokines was confirmed ex vivo in a subgroup of PsA patients (*n =* 150) enrolled in the biomarkers analysis of the phase III randomized clinical trial Psoriatic Arthritis Long-term Assessment of Clinical Efficacy (PALACE) I. At week 24, patients treated with apremilast showed a significantly greater reduction of TNF, IL-6, IL-8, macrophage chemoattractant protein (MCP-1), and macrophage inflammatory protein (MIP1α) serum levels compared to patients who were randomized in the placebo arm; at week 40, serum levels of IL-6, IL-17, IL-23, and ferritin significantly decreased compared to baseline values, whereas IL-10 and IL-1RA significantly increased [52]. While the effect of apremilast on B cells and Immunoglobulin G (IgG) production is modest, it affects the release of pro-inflammatory cytokines from different T cells subset including Th1 (IFN-γ, TNF, and Granulocyte-Macrophage Colony-Stimulating Factor (GM-CSF)), Th2 (Il-5, IL-10, and Il-13) and Th17 (IL-17) cytokines [53]. Despite the inhibitory effect on Th1, Th2, and Th17 for the cytokine production, apremilast does not affect T-cell or B-cell clonal expansion.

In a small study on patients with PsA (*n =* 20) and psoriasis (PsO) (*n =* 30), Mavropoulos et al. investigated the effect of apremilast on IL-10 producing B cells, as these B regulatory (Breg) cells play a crucial role in the balance between regulatory and inflammatory (effector) T cells [54]. Apremilast increased the number of IL-10 producing Breg cells both in PsA and PsO patients, and the number of Bregs inversely correlated with articular and skin clinical scores [47]. On the contrary, apremilast significantly decreased the number of Th1, Natural Killer T cells (NKT), and Th17 cells. Number of Breg cells inversely correlated with IFN-γ^−^, but not IL17^−^ producing T cells and IFN-γ^+^ NKT cells [54].

In a mouse model of collagen induced arthritis (CIA), apremilast delayed the onset of arthritis when administered two weeks after mice immunization; moreover, in mice treated with the PDE4 inhibitor, synovitis, synovial hyperplasia and erosion of bone and cartilage significantly decreased [55]. Moreover, apremilast decreased Th17 and Th1 cells from draining lymph nodes without affecting the number of Treg [55]. In vitro, natural Treg pre-treated with apremilast had a higher Foxp3 and lower IL-17A expression compared to cells treated with dimethyl sulfoxide and maintained their ability to suppress T cell proliferation after stimulation with IL-6 [55]. These results support a role for apremilast in balancing Treg and T effector cells. In the air pouch model, an in vivo model mimicking the synovial cavity with a cell infiltrate composed mostly by neutrophils with a small amount of CD3^+^ T cells, apremilast was shown to decrease significantly the number of neutrophils and TNF production without affecting the IL-1α, IL-10, and IL-6 levels [56].

The effect of apremilast on synovial inflammation and bone homeostasis was investigated ex vivo in patients with Rheumatoid Arthritis (RA) and PsA. In cell cultures of synovial fluid mononuclear cells (SFMC) (mainly composed of lymphocytes and monocytes), apremilast significantly decrease the production of IL-12/IL-23p40 whereas the production of the regulatory cytokine IL-10 was increased [57]. The production of matrix metalloprotease 3 (MMP3) from synovial fibroblast was also inhibited, suggesting that apremilast may also modulate fibrosis. To further investigate the role of apremilast on bone damage, the effect on osteoclastogenesis was investigated. Apremilast did not affect the development of TRAP^+^ osteoclasts in culture of SFMC; however, when osteoclast precursors were cultured on a synthetic inorganic bone-mimetic surface and stimulated with RANKL, the addition of apremilast significantly inhibited the pit formation. On the contrary apremilast did not affect the osteoblast mineralization [57]. Therefore, it is unclear whether apremilast effect on structural damage is only a consequence of its anti-inflammatory effect or may also have a more direct effect on osteoclastogenesis.

### 2.3. RCTs of Apremilast in PsA and IBD

Data from a double-blind, phase 2 trial on 170 adults with active UC treated with apremilast have been published last year by Danese et al. [29]. Patients were naïve to biologic therapy and had failed or were intolerant to conventional immunosuppressants. Patients were randomized to receive apremilast 30 mg (*n =* 57), apremilast 40 mg (*n =* 55), or placebo (*n =* 58) twice daily for 12 weeks; then, patients were randomly assigned to received apremilast 20 mg or 40 mg for additional 40 weeks. The primary endpoint was clinical remission at week 12, defined as a total Mayo score of two or less, with no individual subscore above one. In addition to clinical examination, blood and fecal samples analysis, patients underwent endoscopies and biopsies at week 12 and at week 52.

Although the primary endpoint of clinical remission was not met, at week 12 clinical remission was achieved by a significantly higher percentage of patients in the 30 mg apremilast group than the placebo group (31.6 vs 12.1; *p = 0.01*); at week 52, a further improvement in clinical remission (40.4) was observed in the apremilast 30 mg group. At week 12, both apremilast groups showed greater median percent reductions in inflammatory markers such as C-reactive protein and fecal calprotectin than the placebo group. Surprisingly, a lower percentage of patients receiving apremilast 40 mg twice daily, reached clinical remission, both at week 12 and 52, than apremilast 30mg twice daily (21.8 vs. 31.6 and 32.7 vs. 40.4). This result was mainly attributable to differences in endoscopic improvement, considering that both groups had similar improvements from baseline in clinical Mayo score components (stool frequency score, rectal bleeding score, physician’s global assessment). A similar percentage of patients in the placebo or apremilast 30 mg and 40 mg arms reported at least one adverse event (53.4%, 49.1%, and 63.6%, respectively; *p = ns*). The most frequent apremilast-associated adverse events were headache and nausea. Diarrhea was reported by more placebo than apremilast 30 mg and 40 mg patients (3.4% vs. 1.8% and 0.0%, respectively). The rate of serious adverse events was similar in placebo and apremilast groups, occurring in two patients (3.4%) receiving placebo and one (1.8%) apremilast 40 mg [29].

The clinical development program with apremilast in PsA included 4 phase III RCTs, each enrolling approximately 500 patients: PALACE studies I, II, III, and IV [17,18,19,20,24,25,26,27]. The first 3 shared similar design: 24 weeks, placebo-controlled studies assessing the efficacy and safety of apremilast 20 mg and 30 mg twice a day in patients with PsA with inadequate response to csDMARDs or bDMARDs. PALACE III included patients with at least one qualifying PsO lesion ≥2 cm. The primary end point was the percentage of patients achieving an American College of Reumatology (ACR) 20 response at week 16: significantly more patients treated with apremilast 20 mg and 30 mg achieved the ACR20 compared to placebo; the response rate was higher in bDMARDs naïve patients, but similar in oligo and polyarticular disease subsets [24,26,27]. Overall, 55.3% of patients receiving apremilast 30 mg achieved an ACR20 response at week 52 [58]. In PALACE I, the percentage of patients achieving ACR50 and ACR70 responses at week 24 was significantly higher in the apremilast-treated patients compared with those in the placebo group [58]. The post-hoc analysis of PALACE I–III showed that twice as many patients with moderate disease activity at baseline achieved treatment targets—clinical Disease Activity index of Psoriatic Arthritis (cDAPSA) remission or low disease activity—compared with patients with baseline high disease activity [52,59]. A partial response, ≥30% in cDAPSA improvement, at week 16 was associated to a greater probability to achieve the target by week 52 and patients who already achieved the target at week 16 were likely to remain at target at week 52 [59].

In the long-term observation, at week 260, 67.2%, 44.4%, and 27.4% of patients who continued apremilast achieved ACR20, ACR50, and ACR70 response, respectively [58].

PALACE IV deserves a separate discussion, including patients naïve to csDMARDs and bDMARDs with a shorter disease duration (3.4 years compared to ~7.5 years of the other PALACE studies) who were treated with apremilast monotherapy as first line disease modifying agent [20]. The primary endpoint was set at 16 weeks of treatment: a significantly greater percentage of patients treated with apremilast 20 mg or 30 mg twice daily achieved the ACR20 response. Apremilast was superior to placebo in other efficacy measures including ACR50, ACR70, and change in DAS28 and CDAI [25].

In the ACTIVE study, a phase IIIB RCT, apremilast monotherapy showed early and sustained effect on peripheral arthritis and enthesitis in 219 biological-naïve patients with active PsA; the difference between apremnilast and placebo in ACR20 response rate was significantly different already after 2 weeks of treatment [60].

Extra-articular involvement was not mandatory to be included in the PALACE trials; thus, a separate analysis was made to evaluate the efficacy of apremilast on enthesitis and dactylitis, two domains that greatly contribute to the burden of psoriatic disease perceived by patients. The pooled, post-hoc analysis of patients enrolled in the PALACE I–II–III studies included 945 patients with enthesitis and 633 with dactylitis at baseline. At week 24, mean change in the Maastricht Ankylosing Spondylitis Enthesitis Score (MASES) was significantly higher for apremilast 30 mg compared to placebo; the percentage of patients with a MASES of 0 at week 24 was 22.5% and 27.5% of patients receiving apremilast 20 mg and 30 mg twice a day, respectively; the percentage increase to 55% after 156 weeks of treatment [61].

Mean dactylitis count was significantly improved from baseline in patients treated with apremilast 30 mg but the change in dactylitis was only numerically higher in the apremilast groups; in those patients continuing apremilast through three years, mean improvement in dactylitis count was sustained; at week 156, the percentage of patients with dactylitis count of 0 was 79.6% and 73.9% of patients treated with apremilast 20 mg and 30 mg, respectively [53]. Moreover, the rate of onset of enthesitis and dactylitis in patients with count of 0 at baseline was two-fold higher among patients in the placebo compared to apremilast groups [61].

Overall, 12–14% of patients enrolled in the phase III studies on PsA showed skin involvement, and half of them had a Body Surface Area > 3%; the effect of apremilast also covers skin manifestations with a higher percentage of patients reaching the PASI50 and PASI75 response across the four PALACE studies respect to placebo. In particular, up to 40% in the apremilast 30 mg bid achieved a PASI50 and more than 20% a PASI75.

The effect of apremilast also covers skin manifestation with a greater percentage of patients reaching the PASI50 and PASI75 response across the four PALACE studies [24,25,26,27].

The effect of apremilast on the different domains of PsA reflects on physical function, as demonstrated by the significant improvements in the Health Assessment Questionnaire–Disability Index [24,25,26,27].

The pooled data on 1493 patients who received at least one dose of the study drug in the PALACE I–II–III studies reported the safety profile of apremilast. In the placebo-controlled phase, the percentage of patients in the placebo or apremilast 20 mg and 30 mg arms reporting at least one adverse event was not statistically different (47.8%, 60.5%, 61.5%, respectively); the rate of serious adverse events and the percentage of patients who withdrew from the study due to adverse events were also similar across the treated groups [59]. The rate of serious infection was 0.4% in the placebo group and 0.6% in the apremilast 30 mg twice a day group; no case of tuberculosis reactivation was reported. Exposure-adjusted incidence rates (EAIR) was low during the first 52 weeks (0.2/100 patients-year) and remain stable until week 156 (0.4/100 and 0.5/100 patients-year from week 0 to 104 and 156, respectively) [52]. Rates of malignancies were low across the apremilast dose throughout the 156-week exposure period [59].

Adverse event reported by >5% of patients in any treatment group were diarrhea, nausea, headache, and upper respiratory tract infections; during the weeks 0–52, the most frequent adverse events reported by patients treated with apremilast were diarrhea (13.9%), nausea (12.3%), and headache (9.4%); Gastrointestinal events (diarrhea and nausea) were more frequent in the first 2 weeks of exposure and resolved in 30 days in those patients continuing the treatment. The rate of these adverse events decreased after week 52, until week 156 [59].

### 2.4. Real World Data of Apremilast in PsA

Data on the overall effectiveness of apremilast in PsA patients treated in real-life are still limited (Table 1).

The first report described 71 patients, of whom 61 reached 6 months of follow-up. A half of patients were considered responders based on clinical judgement; those patients showing disease improvement had a shorter disease duration and had failed a lower number of previous drugs [62]. Among the 71 patients, 32 received apremilast in combination with csDMARDs (24 patients treated with dual or triple therapy) or even bDMARDs (9 patients, of whom 6 were treated with apremilast plus TNF inhibitors); the effectiveness of apremilast was similar in the two groups, and the most reported side effects were diarrhea, nausea, and headache [63]. The effectiveness of apremilast add on biological drugs was confirmed in the retrospective analysis of 22 PsA patients, described by Metyas et al., showing a good safety profile of the combination therapy with 6 patients developing adverse event (mainly gastrointestinal) not leading to treatment discontinuation [64].

These initial reports begun to shed light on the possible use of apremilast in real life in refractory patients, suggesting its efficacy and, most of all, its safety as add-on, rescue therapy in patients with inadequate response to bDMARDs. On the other hand, apremilast seems to be an effective option as monotherapy. Among 150 patients starting PsA treatment as monotherapy (34 starting apremilast, 15 methotrexate, and 101 a bDMARD) included in the CORRONA PsA/SpA Registry, those patients treated with apremilast showed a refractory oligoarticular disease and higher disease activity as well as higher scores in Patients Reported Outcomes (PROs) at baseline; after 6 months, the response to apremilast—as assessed by cDAPSA and number of tender and swollen joints—was similar to that observed with bDMARDs and greater compared to methotrexate [65]. The same prescription pattern, predominant oligoarticular involvement, was described by Favalli et al. who carried out a retrospective analysis of 131 PsA patients included in a multicentric study [66]. Other determinant factors for apremilast prescription were enthesitis, mild skin involvement, low risk of damage progression, comorbidities, and contraindications to bDMARDs [58]. Infectious risk, cardiovascular comorbidity and history/concomitant malignancy seems to be the main driver of apremilast administration [66,73].

The clinical benefit of apremilast was confirmed by ultrasonography (US) in a small cohort of patients showing a fast and significant reduction of US inflammatory score already after six weeks of treatment [67]. In a larger cohort, the same authors confirmed the rapid and persistent decrease of US synovial score, significant compared to the baseline evaluation already after 6 weeks and lasting up to 24 weeks; moreover, a significant improvement in tenosynovitis score was also identified at 6 weeks, persisting throughout the follow-up [68]. A similar prompt response was shown on skin lesions: in the cohort of 96 PsA patients described by Balato et al., PASI and Body Surface Area (BSA) started improving as soon as 4 weeks after the beginning of treatment with apremilast and reached the best clinical outcome at week 24 that persisted at week 52 [69]. PASI responses in PsA patients treated in real life are similar to that observed in patients with PsO and seems to be greater compared to those reported in the phase III Efficacy and Safety Trial Evaluating the Effects of Apremilast in Moderate Psoriasis (ESTEEM) studies.

Twelve-months apremilast survival rate in real-life setting ranges from 42.5% to 53.4% [74,75]. In two studies comparing bDMARDs-naive PsA patients starting either apremilast or a bDMARD, the treatment persistence was similar in the two groups, being nearly 43% at 12 months for apremilast [75,76]. Main reasons for discontinuation were lack of efficacy, in a percentage of patients ranging from 5.4% to 46.6%, and side effect, leading to treatment withdrawal in 11.5–26.9% of patients [69,74,76]. As already reported in RCTs, the most frequent side effects were diarrhea, nausea, and headache [69,76]. The safety profile seems to be one of the main drivers in the choice of apremilast. Real life evidence supports the low risk of infections emerged in RCTs. In a large cohort study on patients with PsA and PsO treated with csDMARDs, bDMARDs, or apremilast either as monotherapy or in combination, the incidence of hepatitis C and tuberculosis was low among the 10,000 patients treated with apremilast compared to those treated with DMARDs; moreover, compared to the other drugs, apremilast was associated to the lowest risk of varicella zoster virus infection or reactivation [70]. A single case report showed the safety of apremilast in a patient with PsA and concurrent infections by HIV and HBV.

Risk of major cardiovascular events is another potential safety issue in patients with PsA. In a large cohort study on patients treated with apremilast alone or in combination with either csDMARDs or bDMARDs, for an overall exposure of 7315 person-years, no excess of major cardiovascular events (myocardial infarction, stroke or revascularization) was seen among apremilast-treated patients compared to those treated with csDMARDS [77]. The encouraging results support the relative cardiovascular safety of apremilast; however, rates on myocardial infarction and stroke was higher in patients taking apremilast than in TNF inhibitors monotherapies [77].

## 3. Conclusions

PDEs are a heterogeneous and large family of enzymes catalyzing the degradation of cAMP and cGMP; among PDEs, PDE4 is the most well characterized. It is inhibition elevates intracellular cAMP levels, reducing the expression of inflammatory cytokines among which TNF, IL-17, IFN- γ, IL-23, while increasing regulatory cytokines, such as IL-10. Therefore, PDE4 inhibition has been evaluated as a therapeutic target in the treatment of different chronic inflammatory conditions such as PsA and IBD.

PsA share clinical, genetic, and pathogenic features with IBD, and eSpA represent a frequent clinical evidence of the overlap between gut and joint diseases.

Current therapeutic options in PsA patients and underlying UC are limited to csDMARDs and anti-TNF agents. Apremilast is a PDE4 inhibitor that demonstrated efficacy and a good safety profile in PsA patients both in RCTs and in real-life settings; it also led to a significant improvement in clinical and endoscopic features in UC patients in a phase II RCT.

In view of the above data, the use of apremilast in patients with eSpA seems to be a valuable therapeutic option that should not be overlooked.

## Figures and Tables

**Figure 1 ijms-22-02638-f001:**
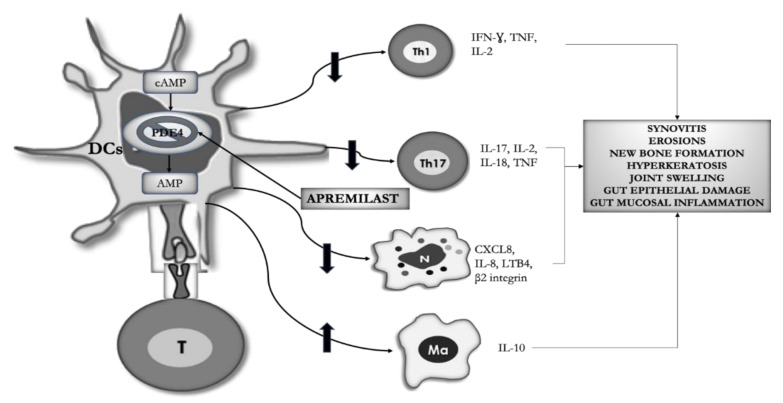
Immunologic rationale for PDE4 inhibition in psoriatic arthritis (PsA) and ulcerative colitis (UC). By blocking PDE4, apremilast increases the intracellular concentration of cAMP leading to an anti-inflammatory effect on different cell types of innate and adaptive immunity, involved in the pathogenesis of the characteristic features of psoriatic arthritis and ulcerative colitis. DCs = Dendritic cells, T = T cell, Th = T helper, IL= interleukin, IFNγ = interferon γ, Ma = macrophage, N = neutrophil, PDE4 = Phosphodiesterase4, cAMP = cyclic adenosine monophosphate.

**Table 1 ijms-22-02638-t001:** Real-world efficacy data on apremilast in patients with Psoriatic Arthritis.

References	N° Patients	Main Findings
Abignano [62]	71	Shorter disease duration and lower exposure to previous bDMARDS in responders vs. non responders patients
Abignano [63]	71	No difference in adverse events between monotherapy (39 patients, 54.9%) and combination therapy (32 patients, 45.1%)
Metyas [64]	22	Apremilast was well tolerated when administered in combination with bDMARDs in patients with inadequate response to bDMARDs
Ogdie [65]	150	Apremilast monotherapy was effective in patients with oligoarticular PsA, more similar to bDMARDs monotherapy than to methotrexate.
Favalli [66]	131	Overall retention: 72.1% and 56.9% at 3 and 6 months. DAPSA minor, moderate, and major response at 3 months: 32.5%, 16.8%, and 6.7%. DAPSA minor, moderate, and major response at 6 months: 22.4%, 17.2%, and 10.3%. DAPSA LDA and remission: 26.9% and 13.4% at 3 months, and 15.5% and 13.7% at 6 months. Male sex and no previous bDMARDs exposure were predictors of DAPSA remission/LDA at 3 months.
Ceccarelli [67]	13	Significant reduction of US inflammatory score after 45 days of treatment
Ceccarelli [68]	34	Apremilast induced an early (6 weeks) and sustained (24) improvement of US inflammatory articular and peri-articular scores
Balato [69]	96	32.3% of patients discontinued apremilast due to lack of efficacy (15.6%), loss of efficacy (4.2%), or adverse events (11.5%). PASI75 at week 4, 12, 24, and 52: 12.5%, 36.5%, 59.4%, and 77.1%
Hagberg [70]	138(27 PsA)	Overall survival rate at 12 months of 53.4%. Reason for discontinuation of apremilast included loss of efficacy (46.4%), adverse events (26.9%), patients’ choice (12.5%), and symptom relief (5.4%).
Mazzilli [71]	113 PsO^+^/^−^ PsA	Better PsO response to apremilast in diabetic patients; reduction of cholesterol and glucose levels after treatment.
Venerito [72]	27	Significant decrease of DAPSA scores after 6 months of treatment; 67.9% of patients achieved LDA or remission. Significant decrease of LEI score at 6 months. Discontinuation in 11.1% of patients for inefficacy. Baseline serum sCD40L level was an independent predictor of DAPSA minor response at 6 months.

bDMARD = biological Disease Modifying Anti-Rheumatic Drugs, DAPSA = Disease Activity in Psoriatic Arthritis, LEI=Leeds Enthesis Index, LDA = low disease activity; PASI= psoriasis area severity index, PsA = psoriatic arthritis, PsO = psoriasis.

## Data Availability

Not applicable.

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
