# Peer review of "Inhibition of Phosphodiesterase-4 in Psoriatic Arthritis and Inflammatory Bowel Diseases"

_ijms, 2021, doi:10.3390/ijms22052638_

Round 1
Reviewer 1 Report
It is an interesting review which might be published in IJMS but somme corrections are needed . I would like to have one more section about psoriasis and skin involvment It would be niece to add some words about ski-jont-gut axis . More over the table including the most important data on the subject should be added.
Author Response
It is an interesting review which might be published in IJMS but some corrections are needed . I would like to have one more section about psoriasis and skin involvement. It would be niece to add some words about ski-jont-gut axis More over the table including the most important data on the subject should be added.
We thank the reviewer for the positive comments on our work.
We intended to focus this narrative review on the link between gut and joint suggesting possible common treatment options. Instead of adding a separate section on psoriasis - out of the scope of the review - we added more on the effect of apremilast on skin manifestation in patients with psoriatic arthritis.
As suggested, the involvement of the skin in the gut-joint axis, from an immunologic point of view, has been integrated (lines 55-67).
We also added a Table summarizing the main findings of the real-life study with apremilast in psoriatic arthritis patients.
Reviewer 2 Report
The article is a very interesting mini-review on the characteristics of apremilast and its use in inflammatory diseases such as psoriatic arthritis and inflammatory bowel diseases; I have some queries:
I think a material and methods section is necessary for two reasons:
It would give more importance to the work (also adding a flow chart would not be a bad idea), and, doing quick research on PubMed, I see some other studies on real-life experiences just published that have been not added to this review. Indicating when the bibliographical research was performed would be in my opinion helpful.
page 2 line 53-54 :"SpA and IBD are closely interconnected, sharing some genetic predisposition (HLA B27), dysregulation of immunologic pathways and inflammatory cytokines such as the IL-23/IL-17 axis and the TNF, and overlapping clinical features. " this sentence needs some references; I suggest you this one doi: 10.1371/journal.pone.0241575. and doi: 10.1080/03007995.2020.1786681.
Thank You
Author Response
The article is a very interesting mini-review on the characteristics of apremilast and its use in inflammatory diseases such as psoriatic arthritis and inflammatory bowel diseases; I have some queries: I think a material and methods section is necessary for two reasons: It would give more importance to the work (also adding a flow chart would not be a bad idea), and, doing quick research on PubMed, I see some other studies on real-life experiences just published that have been not added to this review. Indicating when the bibliographical research was performed would be in my opinion helpful.
Page 2 line 53-54 :"SpA and IBD are closely interconnected, sharing some genetic predisposition (HLA B27), dysregulation of immunologic pathways and inflammatory cytokines such as the IL-23/IL-17 axis and the TNF, and overlapping clinical features. " this sentence needs some references; I suggest you this one doi: 10.1371/journal.pone.0241575. and doi: 10.1080/03007995.2020.1786681.
We sincerely thank the reviewer for its suggestions that offer us the opportunity to improve the quality of our manuscript.
In the revised version of the review we provided a “Literature Search Strategy” including the last update of the literature review. We preferred to report the Search Strategy avoiding a figure not to give the wrong idea of having conducted a systematic review of the literature. As indicated in the literature search, we did not include “psoriasis” since we intended to provide the reader with some suggestions on the gut-joint link and possible common treatment options.
We updated the PubMed search at February 24th but we did not find recent paper evaluating the effectiveness of apremilast in patients diagnosed with psoriatic arthritis; the only recent study not included in our review is the one by Gottlieb et al, reporting the effect of apremilast on skin manifestation in patients with psoriasis (half with concomitant arthritis) from CORRONA registry; we did not include this study since data on joint involvement are lacking. If the referee is aware of any recent study on apremilast effectiveness in patients with PsA we would be glad to extend our discussion.
The suggested reference has been added.
Round 2
Reviewer 2 Report
The authors responded to all queries. The article is in my opinion publishable.